# Drone Based RGBT Tracking with Dual-Feature Aggregation Network

Zhinan Gao [1], Dongdong Li [1,*], Gongjian Wen [1], Yangliu Kuai [2] and Rui Chen [1]

1   College of Electronic Science and Technology, National University of Defense Technology, Changsha 410073, China; gaozhinan22@nudt.edu.cn (Z.G.); wengongjian@sina.com (G.W.); 2019302130162@whu.edu.cn (R.C.)
2   College of Intelligent Science and Technology, National University of Defense Technology, Changsha 410073, China; kuaiyangliu09@nudt.edu.cn
*   Correspondence: lidongdong12@nudt.edu.cn

**Abstract:** In the field of drone-based object tracking, utilization of the infrared modality can improve the robustness of the tracker in scenes with severe illumination change and occlusions and expand the applicable scene of the drone object tracking task. Inspired by the great achievements of Transformer structure in the field of RGB object tracking, we design a dual-modality object tracking network based on Transformer. To better address the problem of visible-infrared information fusion, we propose a Dual-Feature Aggregation Network that utilizes attention mechanisms in both spatial and channel dimensions to aggregate heterogeneous modality feature information. The proposed algorithm has achieved better performance by comparing with the mainstream algorithms in the drone-based dual-modality object tracking dataset VTUAV. Additionally, the algorithm is lightweight and can be easily deployed and executed on a drone edge computing platform. In summary, the proposed algorithm is mainly applicable to the field of drone dual-modality object tracking and the algorithm is optimized so that it can be deployed on the drone edge computing platform. The effectiveness of the algorithm is proved by experiments and the scope of drone object tracking is extended effectively.

**Keywords:** RGBT tracking; Drone based object tracking; transformer; feature aggregation





## 1. Introduction

Object tracking is one of the fundamental tasks in computer vision and has been widely used in robot vision, video analysis, autonomous driving and other fields [1]. Among them, the drone scene is an important application scenario for object tracking which assist drones in playing a crucial role in urban governance, forest fire protection, traffic management, and other fields. Given the initial position of a target, object tracking is to capture the target in subsequent video frames. Thanks to the availability of large datasets of visible images [2], visible-based object tracking algorithms have made significant progress and achieved state-of-the-art results in recent years. Currently, due to the diversification of drone missions, visible object tracking is unable to meet the diverse needs of drones in various application scenarios [3]. Due to the limitations of visible imaging mechanisms, object tracking heavily relies on optimal optical conditions. However, in realistic drone scenarios, UAVs are required to perform object tracking tasks in dark and foggy environments. In such situations, visible imaging conditions are inadequate, resulting in significantly noisy images. Consequently, object tracking algorithms based on visible imaging fail to function properly.

Infrared images are produced by measuring the heat emitted by objects. Compared with visible images, infrared images have relatively poor visual effects and complementary target location information [4,5]. In addition, infrared images are not sensitive to changes in scene brightness, and thus maintain good imaging results even in poor lightning environments. However, the imaging quality of infrared images is poor and the spatial resolution

and grayscale dynamic range are limited, resulting in a lack of details and texture information in the images. In contrast, visible images are very rich in details and texture features. In summary, visible and infrared object tracking has received increasing attention as it can meet the mission requirements of drones in various scenarios, due to the complementary advantages of infrared and visible images (Figure 1).

Currently, two main kinds of methods in visual object tracking are deep learning (DL)-based methods and correlation filter (CF)-based approaches [1]. The methods based on correlation filtering utilize Fast Fourier Transform (FFT) to perform correlation operation in the frequency domain, which have a very fast processing speed and run in real-time. However, their accuracy and robustness are poor. The methods based on neural network mainly utilize the powerful feature extraction ability of neural network. Their accuracy is better than that of correlation filtering based methods while their speed is slower. With the proposal of Siamese networks [6,7], the speed of neural network-based tracking methods has been greatly improved. In recent years, the neural network-based algorithm has become the mainstream method for object tracking.

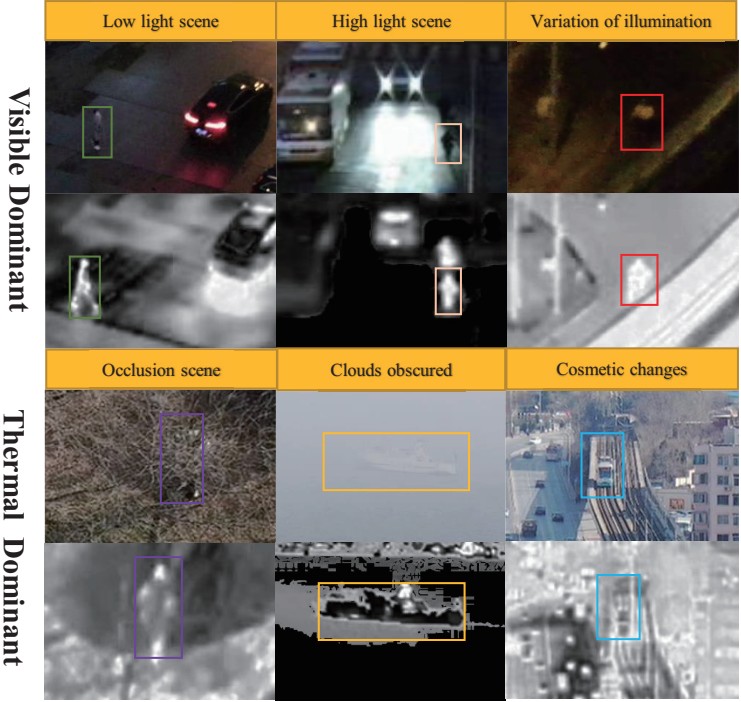

**Figure 1.** These are some visible-infrared image pairs captured by drones. In some scenarios, visible images may be difficult to distinguish different objects, while infrared images can continue to work in these scenarios. Therefore, information from visible and infrared modalities can complement each other in these scenarios. Introducing information from the infrared modality is very beneficial for achieving comprehensive object tracking in drone missions.

The transformer structure has achieved great success in the field of computer vision [8]. By introducing long-range attention mechanisms, it alleviates the problem of limited receptive fields in CNNs and has achieved State-of-the-art (SOTA) performance in multiple tasks in the field of computer vision [9,10]. Inspired by the remarkable success of transformer-based Siamese networks in object tracking on visible images, we propose a similar approach using a transformer-based Siamese network for RGB-Thermal (RGBT) tracking. Additionally, we drew inspiration from the transformer structure and used an attention mechanism to fuse visible and infrared image features in both spatial and channel dimensions. We visualized the effectiveness of this module through heat maps and achieved promising results on the drone dataset VTUAV [11] and drone hardware platform. The main contributions can be summarized as follows:

- A lightweight network applied to a drone visible-infrared object tracking mission is porposed with Swin transformer as the backbone network.
- A Dual-Feature aggregation module is integrated into this network, which aggregates visible-infrared image features from both spatial and channel dimensions using attention mechanisms. Ablation experiments are conducted to vefify the effectiveness of this module in fusing two modalities.
- Extensive experiments are conducted on the VTUAV dataset and drone hardware platform, the results showed that our tracker achieved good performance compared with other mainstream trackers.

## 2. Related Works

### 2.1. RGBT Tracking Algorithms

Many RGBT trackers have been proposed so far [12–15]. Due to the rapid development of RGB trackers, current RGBT trackers mainly consider the problem of dual-modal information fusion within mature trackers finetuned on the RGBT tracking task, where the key is to fuse visible and infrared image information. Several fusion methods are proposed, which are categorized as image fusion, feature fusion and decision fusion. For image fusion, the mainstream approach is to fusion image pixels based on weights [16,17], but the main information extracted from image fusion is the homogeneous information of the image pairs, and the ability to extract heterogeneous information from infrared-visible image pairs is not strong. At the same time, image fusion has certain requirements for registration between image pairs, which can lead to cumulative errors and affect tracking performance. Most trackers aggregate the representation by fusing features [18,19]. Feature fusion is a more advanced semantic fusion compared with image fusion. There are many ways to fuse features, but the most common way is to aggregate features using weighting. Feature fusion has the potential of high flexibility and can be trained with massive unpaired data, which is well-designed to achieve significant promotion. Decision fusion models each modality independently and the scores are fused to obtain the final candidate. Compared with image fusion and feature fusion, decision fusion is the fusion method on a higher level, which uses all the information from visible and infrared images. However, it is difficult to determine the decision criteria. Luo et al. [12] utilize independent frameworks to track in RGB-T data and then the results are combined by adaptive weighting. Decision fusion avoids the heterogeneity of different modalities and is not sensitive to modality registration. Finally, these fusion methods can also be used complementarily. For example, Zhang [11] used image fusion, feature fusion and decision fusion simultaneously for information fusion and achieved good results in multiple tests.

### 2.2. Transformer

Transformer originates from natural language processing (NLP) for machine translation and has been introduced to vision recently with great potential [8]. Inspired by the success in other fields, researchers have leveraged Transformer for tracking. Briefly, Transformer is an architecture for transforming one sequence into another one with the help of attention-based encoders and decoders. The attention mechanism can determine which parts of the sequence are important, breaking through the receptive field limitation of traditional CNN networks and capturing global information from the input sequence. However, the attention mechanism requires more training data to establish global relationships. Therefore, Transformer will have a lower effect than traditional CNN networks in some tasks with smaller sample size and more emphasis on regional relationships [20]. Additionally, the attention mechanism is able to replace correlation filtering operations in the Siamese network by finding the most relevant region to the template in the search area in a global scope. The method of [9] applies Transformer to enhance and fuse features in the Siamese tracking for performance improvement.

### 2.3. UAV RGB-Infrared Tracking

Currently, there are few visible-light-infrared object tracking algorithms available for drones, mainly due to two reasons. Firstly, there is a lack of training data for visible-light-infrared images of drones. Previously, models were trained using infrared images generated from visible images due to the difficulty in obtaining infrared images. With the emergence of datasets such as LasHeR [21], it is now possible to directly use visible and infrared images for training. In addition, there are also datasets such as GTOT [22], RGBT210 [23], RGBT234 [24], etc. available for evaluating RGBT tracking algorithm performance. However, in the field of RGBT object tracking for drones, only the VTUAV [11] dataset is available. Due to the different imaging perspectives of images captured by drones compared to normal images, training algorithms with other datasets does not yield good results. Secondly, existing algorithms have slow running speeds, making them difficult to use directly. Existing mainstream RGBT object tracking algorithms are based on deep learning, which have to deal with both visible and infrared images at the same time, with a large amount of data, a complex algorithmic structure and a low processing speed,such as JMMAC (4fps) [25], FANet (2fps) [18], MANnet (2fps) [26]. In drone scenarios, there is a high demand for speed in RGBT object tracking algorithms for drones. It is necessary to simplify the algorithm structure and improve its speed.

### 3. Material and Methods

This section introduces our visible-infrared object tracking algorithm. which is inspired by siamese-based RGB object tracking algorithms and adapted to RGBT tracking tasks based on SwinTrack [10]. The network is mainly divided into four parts: Feature Extraction Network, Dual-Feature Aggregation Network, Transformer-based Feature Fusion and Detection Head. The structure of network is shown as Figure 2.

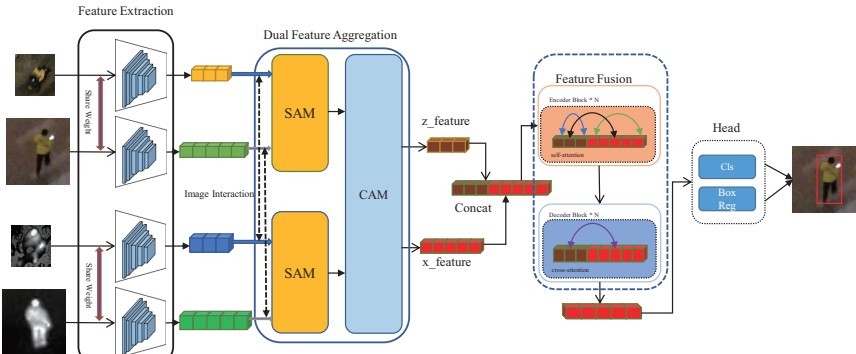

**Figure 2.** Architecture of our Transformer RGBT Tracking framework. This framework contains four fundamental components: Feature Extraction Network,Dual-Feature Aggregation Network, Feature Fusion Network and Detection Head.

### 3.1. Feature Extraction Network.

Traditionally, the ResNet network [27] is commonly used for feature extraction in computer vision tasks. However, with the development of the Transformer in the field of computer vision, feature extraction networks based on the Transformer have achieved better results than ResNet [10]. Moreover, our model is entirely based on the Transformer structure, using Swin-Transformer as the feature extraction network, which can provide a more compact feature representation and richer semantic information. This is very advantageous for the subsequent modules.The structure of Swin-Transformer is shown as Figure 3.

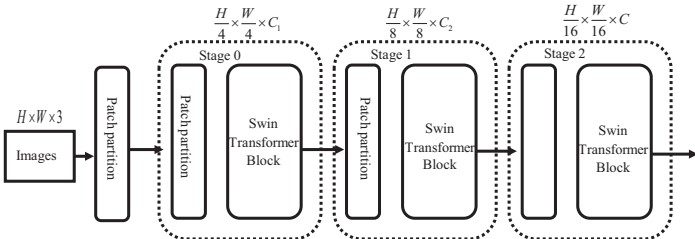

**Figure 3.** Architecture of Swin-Transformer. To accelerate the computation speed, images are divided into patches by Patch partition and fed into the network.

Our tracker follows the Siamese network structure, which requires a pair of image patches as inputs, i.e.,a template image $z \in \mathbf{R}^{H_z \times W_z \times 3}$ and a search region image $x \in \mathbf{R}^{H_x \times W_x \times 3}$. The image pairs are firstly segmented into small patches and fed into the network. Attention operations are performed on these small patches, which significantly reduces the computation cost of the transform. Finally, template tokens $t_z \in \mathbf{R}^{\frac{H_z}{S} \times \frac{W_z}{S} \times C}$ and region tokens $t_x \in \mathbf{R}^{\frac{H_x}{S} \times \frac{W_x}{S} \times C}$ are generated, where $s$ is the stride of the backbone network and $C$ is the hidden dimension of the feature. To reduce model complexity, the lightweight Swin-Transform is used,where $s = 16$ and $C = 386$.

For single-modality tracking, a shared feature extraction network that shares parameters between the template and search images is sufficient. However, our task is RGB-T tracking with different imaging characteristics in each modality. Therefore, here visible and infrared features are extracted separately and two independent feature extraction modules are needed to extract features separately.

### 3.2. Dual-Feature Aggregation Network

Inspired by works such as Convolutional Block Attention Module (CBAM) [28] and Squeeze-and-Excitation Networks (SENet) [29], in the process of fusing visible and infrared information, we used attention mechanisms to enhance useful feature information in both spatial and channel dimensions. We proposed a Dual-Feature aggregation network and its structure diagram is shown in Figure 4.

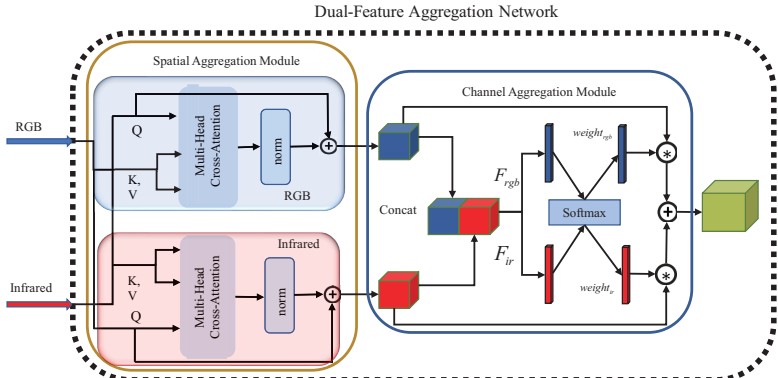

**Figure 4.** Architecture of Dual-Feature Aggregation Network. It mainly consists of two modules, namely Spatial Aggregation Module (SAM) and Channel Aggregation Module (CAM).

Different from other feature fusion methods that use a single modality to enhance another modality, we simultaneously enhance the visible-infrared dual-modality feature information by attention mechanism during the fusion process, which is very useful in some scenarios where single-modality algorithm is limited. The expression of the attention mechanism is:

$$Attention(Q, K, V) = Softmax(\frac{QK^T}{\sqrt{d_k}}V) \tag{1}$$

where $Q$, $K$ and $V$ stand for Query, Key and Value respectively, $\sqrt{d_k}$ means dimensionality normalization.

The attention mechanism is a feature weighting method. By selecting different Query, Key and Value, the image can be searched as a whole and the features of important areas in the image can be enhanced, so that the algorithm can pay more attention to important target areas in the image. For visible-infrared heterogeneous images, although their imaging methods are different, the target information is similar. For dual-modality object tracking, such similar target information is needed. By using the attention mechanism on heterogeneous images, by selecting different values of Query, Key and Value on heterogeneous images, one modality information can be used to enhance another modality information. In this case, the attention mechanism will strengthen the important target region in heterogeneous images and suppress the noise region in each modality. It is more conducive to the fusion of dual-modality features. The Dual-feature aggregation network consists of two modules: Spatial Aggregation Module (SAM) and Channel Aggregation Module (CAM). Below we will introduce these two modules respectively.

The Spatial Aggregation Module mainly focuses on the spatial features of the image. Inspired by the transformer structure, the attention mechanism searches for the regions of interest at any position in the image. By using the attention mechanism with an infrared image as the template and a visible image as the search image, the model can search for the most similar region in the visible image based on the information from the infrared image. This effectively aligns the information from the two modalities and highlights the parts of the visible image that are most similar to the infrared image. This approach allows for the incorporation of information from one modality into another while preserving the original information as much as possible. Similarly, by swapping the inputs of the visible and infrared images, the visible information can be used to calibrate the infrared image, resulting in calibrated features for both the infrared and visible images. This is why this module is called "Dual-feature". This process can be represented as:

$$SA_{RGB} = F_I + (MCA(F_I, Concat_s(F_{RGB}, F_I))) \tag{2}$$

$$SA_I = F_{RGB} + (MCA(F_{RGB}, Concat_s(F_I, F_{RGB}))) \tag{3}$$

where $F_{RGB}$ and $F_I$ are the visible-infrared feature information extracted by the backbone network. $MCA$ is a multi-head Cross-Attention module, $Concat_s$ means concatenate in spatial domain.

The Channel Aggregation Module focuses on the features of image channels. The calibrated visible and infrared image features are obtained after spatial aggregation. If these features are directly fed into the encoder, the channel dimension will be twice that of the original features, which will seriously affect the efficiency of the encoder. Therefore, channel aggregation is needed to select the most important channel features from visible and infrared images, reduce the channel dimension and further fuse the information from visible and infrared images. The process can be expressed as:

$$weight_{RGB} = Softmax(\mathcal{F}_{RGB}(Concat_c(SA_{RGB}, SA_I))) \tag{4}$$

$$weight_I = Softmax(\mathcal{F}_I(Concat_c(SA_{RGB}, SA_I))) \tag{5}$$

$$F_{out} = weight_{RGB} \times SA_{RGB} + weight_I \times SA_I \tag{6}$$

where $SA_{RGB}$ and $SA_I$ represent the features that have been spatially aggregated. $\mathcal{F}_{RGB}$ and $\mathcal{F}_I$ are pooling layers used to generate weight parameters on channels. $Concat_c$ means concatenate in channel domain. $F_{out}$ is the output of the Dual-Feature Aggregation Network.

### 3.3. Transformer-Based Feature Fusion Network

After the Dual-Feature aggregation network, fused template and search features are obtioned. During the fusion process, an encoder and decoder are utilized to fuse the template and search information and the resulting output is then refined. The encoder compresses the template and search features into a more compact representation. The decoder then decodes the compressed features back into the original feature space, achieving feature fusion. In the search regions, we utilize the fused features to locate the region that has the highest correlation with the template in the candidate region. Finally, these regions are fed into the detection head for detection. It is worth noting that we have employed concatenation-based fusion architecture in SwinTrack [10], which significantly reduces the model size and number of parameters compared to traditional methods.

The encoder consists of a sequence of Transformer blocks. Each block contains a Multi-Head Self-Attention (MSA) module and a Feed-Forward Network (FFN). The FFN contains a two-layer Multi-Layer Perceptron (MLP) with a GELU activation layer inserted after the first linear layer. To avoid overfitting, Layer Normalization (LN) is used. Moreover, residual connections are employed in both the MSA and FFN modules to facilitate gradient backpropagation. To reduce model complexity, four layers of Transformer blocks are used here as encoder. The process of encoder can be expressed as: Concatenate the features of template and search:

$$F = Concat(f_z, f_x) \tag{7}$$

Perform Attention operation on concatenated features in each encoder block:

$$F_{MSA} = F + MSA(LN(F)) \tag{8}$$

$$F_{FFN} = MLP(LN(F_{MSA})) + F_{MSA} \tag{9}$$

Separate the concatenated features into their original template and search features:

$$f_z, f_x = DeConcat(F_{FFN}) \tag{10}$$

Here $f_x$ and $f_z$ are respectively the template and search from Dual-Feature Aggregation Network. $MSA$ is a multi-head Self-Attention module.

The decoder is composed of a Multi-Head Cross-Attention (MCA) module and the remaining parts are the same as the encoder and one layer of Transformer block is used here as decoder. The entire process of the decoder can be represented as follows:

$$F_{MCA} = f_x + MCA(LN(f_x), LN(Concat(f_x, f_z))) \tag{11}$$

$$F = MLP(LN(F_{MCA})) + F_{MCA} \tag{12}$$

Here $f_x$ and $f_z$ are produced by *DeConcat* in Encoder module. $F$ will be fed to the Head network to generate a classification response map and a bounding box regression map.

### 3.4. Head and Loss

The Head network is split into two branches: classification and bounding box regression. Both are three-layer MLP networks that receive the feature map output from the decoder and respectively predict the classification response map $R_{cls} \in \mathbb{R}^{H_x \times W_x \times 2}$ and bounding box regression map $R_{reg} \in \mathbb{R}^{H_x \times W_x \times 4}$.

The classification Head receives the feature map output from the decoder and predicts the binary classification results of the search region. Only the annotated box is considered as a positive sample, while the rest are negative samples. As a result, the number of positive and negative samples is imbalanced. To alleviate this issue, we use a hyperparameter $\mu$, which is set to 0.0625 based on experimental results, to reduce the loss from negative

samples by a factor of $\mu$. We use the standard binary cross-entropy loss for classification, which is defined as follows:

$$\mathcal{L}_{cls} = -\sum_i [y_i log(p_i) + \mu(1 - y_i)log(1 - p_i)] \tag{13}$$

Here, $y_i$ denotes the ground-truth label of the i-th sample, $y_i = 1$ denotes foreground and $p_i$ denotes the probability belong to the foreground predicted by the learned model.

For the bounding box regression, we use a linear combination of L1-norm loss $\mathcal{L}_1(.,.)$ and the generalized IoU loss $\mathcal{L}_{Giou}(.,.)$ [30]. The loss is calculated only for positive samples, while negative samples are ignored. The regression loss is defined as follows:

$$\mathcal{L}_{reg} = \sum_i [\lambda_G \mathcal{L}_{Giou}(b_i, \hat{b}) + \lambda_1 \mathcal{L}_1(b_i, \hat{b})] \tag{14}$$

where $\hat{b}$ denotes the normalized ground-truth bounding box. $\lambda_G = 3$ and $\lambda_1 = 4.3$ are hyperparameter weights determined through experiments.

## 4. Results

### 4.1. Implementation Details

**Offline training.** Experments are conducted on the VTUAV dataset and it is worth noting that the VTUAV dataset is annotated every ten frames. Therefore, there are only about 20,000 pairs of accurately usable training samples. To overcome this problem, the Lasher dataset is used for pre-training. The sizes of search region patch and template patch are $224 \times 224$ and $112 \times 112$, respectively. We trained the model using an AdamW optimizer with different initial learning rates for different modules. The backbone network used the Swin Transformer-Tiny pre-trained on Imagenet1K [31], with a stride of 16 and producing a feature map of size $14 \times 14$. The visible backbone network was most compatible with the pre-trained network and was set with a learning rate of $5 \times 10^{-5}$. The infrared backbone network required task fine-tuning, so its learning rate was set to $1 \times 10^{-4}$. The learning rates for all other modules were set to $5 \times 10^{-4}$, with a weight decay of $1 \times 10^{-4}$. Due to the use of concatenation-based Transform structures, our model has much lower GPU memory consumption compared to Transt [9]. We set the batchsize as 40, which can be trained on a single Nvidia Titan RTX GPU. We trained the model for 100 epochs and the learning rate decreased by a factor of 10 after 80 epochs.

**Online Inference.** We follow the inference steps of the Siamese network. First, we initialize the template based on the annotation results of the first frame. The target object is placed in the center of the image with a background area factor of 2. Then, we generate the search region based on the detection results. The background area factor for the search region is 5. During the inference process, the Detection Head outputs a $14 \times 14$ classification response feature map. We use a Hanning window to incorporate the prior information of the target's position into the tracking process, thereby suppressing sudden changes in the target's position. The process can be expressed as:

$$cls = (1 - \gamma) \times r_{cls} + \gamma \times h \tag{15}$$

Here $r_{cls}$ is classification response feature map, $\gamma$ is the weight parameter and $h$ is the Hanning window with the same size as $r_{cls}$. And $\gamma = 0.49$ always get a very good result by experiments. After determining the target position based on the classification response feature map, the target's bounding box is estimated on the position response map. The new search region is then fed into the tracking network, completing a full target tracking inference process.

**Evaluation metrics.** In our experiment, all the trackers are run in one-pass evaluation (OPE) protocol and evaluated by Maximum Success Rate (MSR) and Maximum Precision Rate (MPR), which are widely used in RGB-T tracking [22–24]. A total of 175 short-term tracking video sequences from the VTUAV dataset were used, with each test sequence ranging from 2000 to 15,000 frames. The annotation results were labeled every ten frames, and the final evaluation results were tested using a sampling method.

### 4.2. Ablation Experiment

In order to verify the effectiveness of each component in the network, ablation experiments were conducted on the modules in the network and the results are shown as Table 1 and Figure 5.

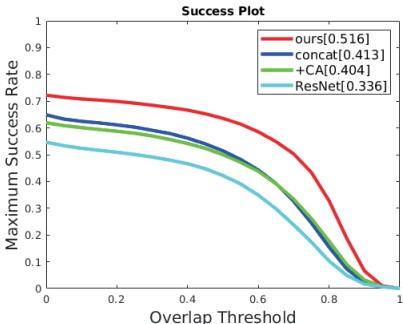 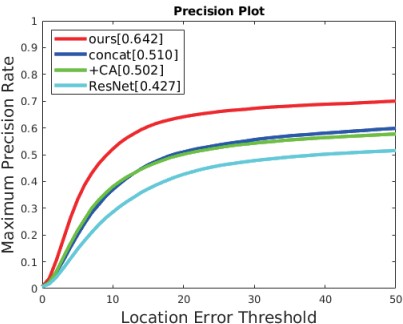

**Figure 5.** Ablation experiment result.

**Table 1.** Comparative Experiment Analysis Table.

| RGBT Tracker | Maximum Success Rate(MSR) | Maximum Precision Rate (MPR) | Parameter |
|---|---|---|---|
| Concat | 41.3 | 51.0 | 41.9 M |
| ResNet | 33.6 | 42.7 | 41.3 M |
| +CAM | 40.4 | 50.2 | 41.6 M |
| Ours | 51.6 | 64.2 | 47.9 M |

Firstly, the network without the Dual-Feature Aggregation Network (Concat) was tested as a baseline on the VTUAV dataset, and its MSR and MPR decreased by 10.3 and 13.2, respectively, which fully demonstrated the effectiveness of the Dual Feature Aggregation Network in RGB-T tracking. Secondly, experiments were conducted by replacing SwinTransform with ResNet as the backbone network (ResNet), and it was found that its performance decreased. This is because the stride of ResNet is 8, which requires the use of smaller template ($56 \times 56$) and search ($112 \times 112$) sizes, thus increasing the problem of limited receptive fields. Based on the baseline network (Concat), directly aggregating the visible and infrared channels after concatenation (+CAM) resulted in a relative decrease of 0.9 and 0.8 compared to the baseline. But performing spatial aggregation before channel aggregation has greatly improved its effectiveness (Ours).

Through analyzing the heatmap (Figure 6), it was found that when directly performing channel aggregation, multiple non-target areas with high confidence scores appeared in the heatmap. This would interfere with the network and easily lose the target, which is the main reason for the performance degradation. Although direct channel aggregation also complete the fusion of dual-modal information, this method has large errors and lead to tracking failure. However, by first using a spatial aggregation module to aggregate visible and infrared channels, the target information in the dual-modal can be selectively retained while suppressing noise. This reflects the effectiveness of the Dual Feature Aggregation Network in preserving bimodal information.

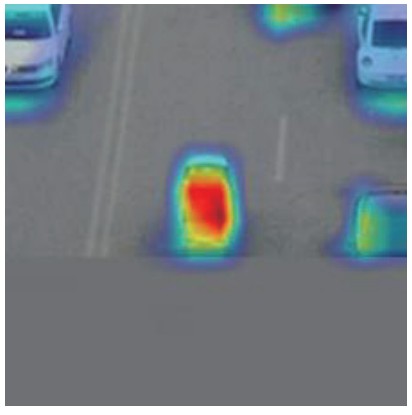 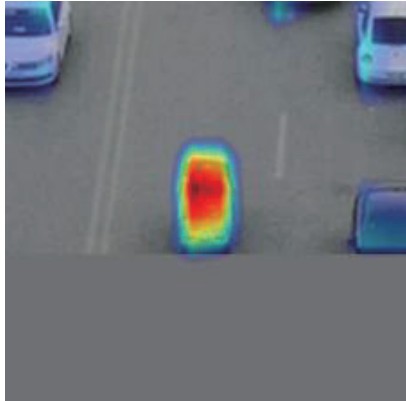

a. +CAM    b. Dual-Feature Aggregation

**Figure 6.** Ablation experiment result.

*4.3. Contrastive Experiment*

Currently, the main mainstream deep-Learnning based RGBT object tracking algorithms are JMMAC [25], FANet [18], MANnet [26], DAFNet [32], ADRNet [14], FSRPN [15], etc., but some of these algorithms are too slow and not applicable to UAV RGBT object tracking tasks. We tested our algorithm on the VTUAV dataset and compared it with three additional fast trackers (DAFNet, ADRNet, FSRPN). Both DAFNet [32] and ADRNet [14] are the best performing multi-domain network trackers. The multi-domain network mainly performs classification and regression tasks in each domain such as visible and infrared and finally obtains the final tracking result of RGBT through competitive learning. FSRPN [15] is a Siamese-based tracker, which uses the pipeline of Siamese to improve the tracking accuracy and speed up the processing speed of the algorithm. The results are shown as Table 2 and Figure 7.

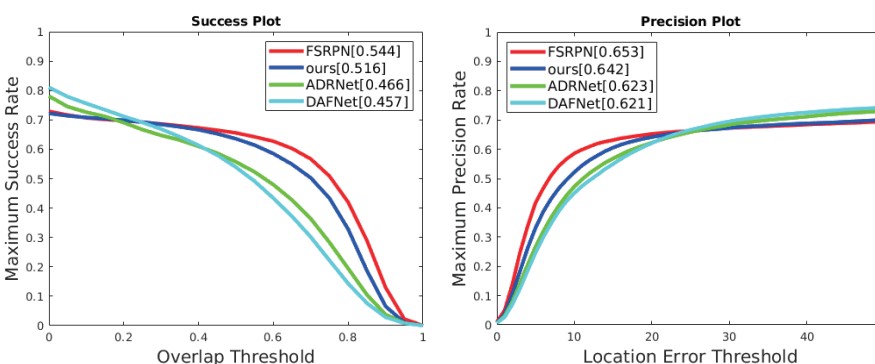

**Figure 7.** Comparison experiment result.

**Table 2.** Comparative Experiment Analysis Table.

| RGBT Tracker | Maximum Success Rate(MSR) | Maximum Precision Rate (MPR) | FPS | Parameter |
|---|---|---|---|---|
| DAFNet | 45.7 | 62.1 | 21.0 | 68.5 M |
| ADRNet | 46.6 | 62.3 | 25.0 | 54.1 M |
| FSRPN | 54.4 | 65.3 | 30.3 | 53.9 M |
| Ours | 51.6 | 64.2 | 31.2 | 47.9 M |

In the comparative experiments, our algorithm outperformed DAFNet and ADRNet in Maximum Success Rate and Maximum Precision Rate (MSR higher than DAFNet by 19% and ADRNet by 17%, MPR higher than DAFNet and ADRNet by 3%), but was slightly

inferior to the FSPRN algorithm. Our algorithm has been specifically designed for drone missions, focusing on simplifying the network structure, reducing the number of network parameters and enhancing algorithm speed. As a result, our algorithm has successfully achieved a favorable balance between performance and efficiency on VTUAV in comparison to mainstream algorithms.

*4.4. Drone Hardware Platforms*

Our algorithms will eventually be deployed on a drone hardware platform, the components of which are described here, and the composition diagram is shown in Figure 8.

The DJI M300 drone is an industry-level unmanned aerial vehicle that can fly for 30 min in the air with a payload of up to 9 kg. It has a maximum flight altitude of 1500 m, making it suitable for most drone scenarios and tasks. The H20T camera is a visible-light and thermal infrared camera that can capture both visible-light and infrared images simultaneously. We use the H20T camera to complete our drone RGBT tracking tasks. We use Nvidia Orin NX as the on-board processing platform with the specific parameters shown in Table 3.

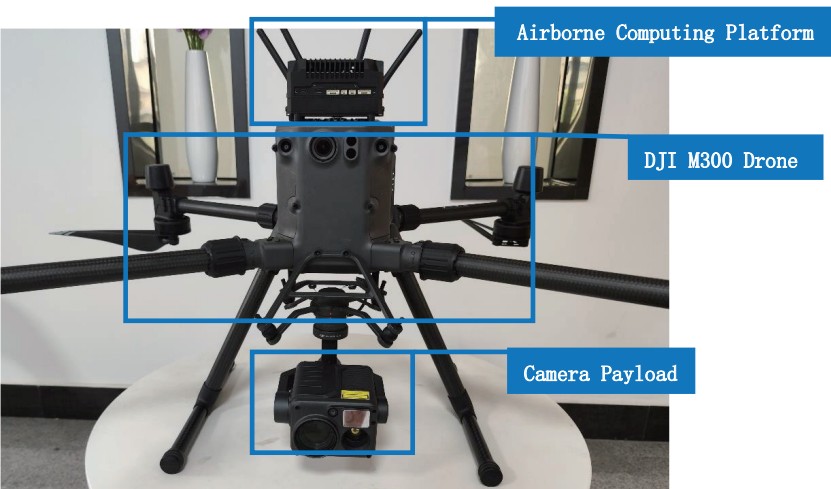

**Figure 8.** Structure of Drone Hardware Platforms. It mainly consists of three components, the DJI M300 drone, the H20T camera load, the Airborne Computing Platform Nvidia Orin NX.

**Table 3.** Nvidia Orin NX parameters Table *.

| CPU | CPU Frequency | Display Memory | Computational Performance |
|---|---|---|---|
| Arm Cortex-A78AE | 2 GHz | 16 GB | 100TOPS |

* Parameters from official Nvidia documentation.

*4.5. Visualization and Analysis*

4.5.1. Algorithmic Effect of Drone Hardware Platform

We tested the images captured by the drone hardware platform with a processing speed of 13.1 fps when running the RGBT object tracking algorithm on the onboard computing platform, and the tracking results are shown in Figure 9.

From the tracking results, it can be observed that the infrared modality can solve the problem of tracking failure under conditions such as occlusion and lighting changes in complex scenes. Therefore, using visible-infrared object tracking can expand the scope of drone object tracking tasks and improve the environmental adaptability of drone object tracking tasks.

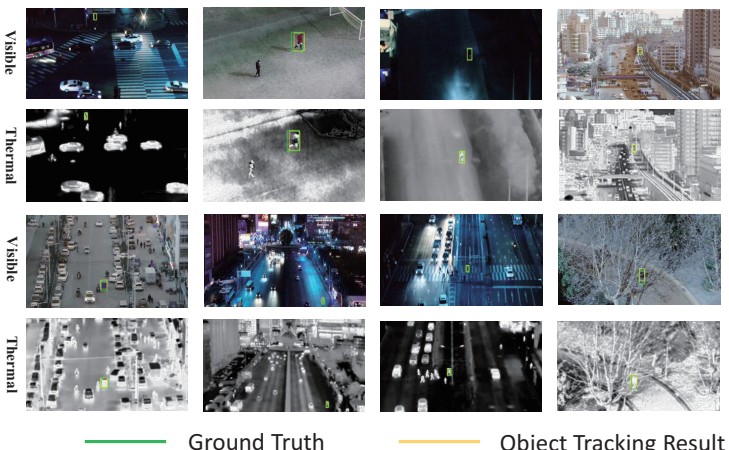

Ground Truth ——— Object Tracking Result

**Figure 9.** Graph of actual results of the algorithm.

### 4.5.2. Heatmap

We present the heatmaps generated from various main modules in the network, as shown in Figure 10. From the heatmaps, it can be observed that the backbone network extracts features separately from the visible and infrared images. Due to the differences between visible and infrared images, the features extracted by the network are also different. These features are then fed into the Dual Feature Aggregation Network, which combines the information from both modalities to obtain a fused feature map. In pedestrian-211 and Tricy-006 sequences, it can be seen more clearly that the aggregated feature map integrates all the features from both images. The fused feature map is then passed through the encoder and decoder modules, which focus the network's attention on the target. Finally, the maximum target response map is fed into the Head for classification and regression.

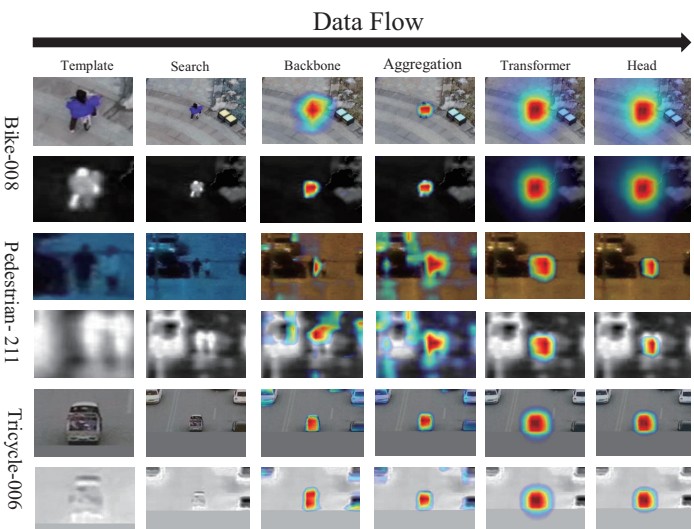

**Figure 10.** Heatmaps of Various Modules in the Network. The left two images are the visible-infrared template and search images input into the network. Following the direction of the network data flow, the heatmaps show the responses of the Backbone network, Feature Aggregation Network, Transform Fusion Network and Head modules, respectively.

### 4.5.3. Typical Failure Cases

We show some typical failure cases of our tracker in Figure 11.

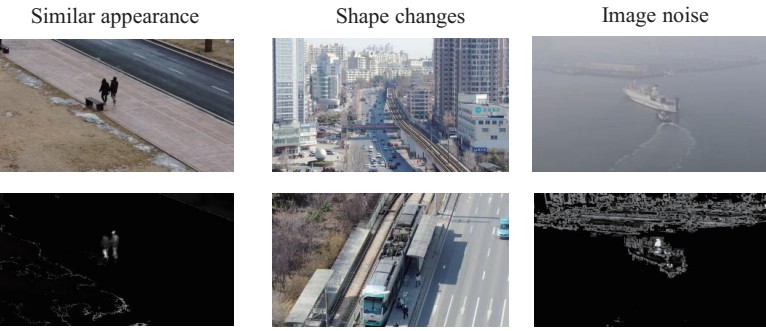

**Figure 11.** Picture of the typical failure cases of our tracker.

The first image shows similar appearance, where the target is interfered with by similar targets in visible and infrared images. The Siamese network lacks the ability to distinguish between similar targets as its core is the matching of templates and searches. Although we introduce prior information about the target position by using the Hanning window and penalize sudden changes in the target's position, tracking similar targets at close range is still prone to failure. The second image shows shape changes. our tracker is non-temporal. When the appearance of the target in the search is significantly different from that in the template, the tracker will loses the target. Thus, temporal information is essential in some long sequence tracking processes. The third image shows imaging noise. In some environments, the infrared images obtained contain noise, severely affecting the quality of infrared images. When the single-mode noise is too strong, the tracker will be affected by interference from noise, affecting the tracking effect.

## 5. Discussion

This study primarily focuses on visible-infrared dual-modality object tracking in drone scenarios. We have conducted ablation experiments and visualization analysis to validate the effectiveness of the proposed dual-feature aggregation network in aggregating visible-infrared modality information. Our algorithm outperforms other mainstream algorithms in terms of tracking accuracy, while utilizing fewer parameters and achieving faster running speeds. Our algorithm is specifically designed for drone scenarios and can be seamlessly deployed and executed on the Nvidia Orin NX, a drone edge computing platform with limited computing resources. To evaluate the algorithm's performance, we conducted tests in an open scene using the aforementioned hardware platform. The results demonstrate that leveraging dual-modality information can significantly enhance the accuracy and robustness of object tracking, particularly in scenarios with illumination changes and occlusions. Additionally, we have analyzed the failure cases encountered during the experiments to identify potential areas for future research. The performance of our algorithm is degraded in scenes with similar appearance, shape changes and image noise. Furthermore, the algorithm's processing speed still falls short of meeting real-time requirements on edge computing devices. These challenges serve as important considerations for future improvements.

## 6. Conclusions

In this work, we mainly designed a visible-infrared object tracking network based on the Transformer architecture. It consists of four components, among which we focused on designing a Dual-Feature Aggregation Network to fuse visible and infrared information. Through ablation experiments and visualization analysis, we demonstrated the effectiveness of the Dual-Feature Aggregation Network. The algorithm is mainly for the task of RGBT object tracking in drone scenarios and the algorithm is simplified so that it can run on the drone edge computing platform. Compared with the mainstream RGBT object tracking algorithms, our algorithm still achieves better performance.

**Author Contributions:** Conceptualization, Z.G. and D.L.; methodology, Z.G.; software, Z.G.; validation, Z.G., D.L. and R.C.; formal analysis, Z.G.; investigation, Z.G.; resources, D.L.; data curation, Z.G.; writing—original draft preparation, Z.G.; writing—review and editing, Z.G., D.L., Y.K.; visualization, Z.G.; supervision, Z.G.; project administration, D.L., G.W.; funding acquisition, D.L. and Y.K. All authors have read and agreed to the published version of the manuscript.

**Funding:** This work is supported by the National Natural Science Foundation of China (NSFC) (NO.62102426), the natural science foundation of Hunan Province (NO.2021JJ40683), the Science and Technology Innovation Plan Project of Hunan Province (NO.2021RC2072) and the scientific research project of National University of Defense Technology (NO. ZK20-47, ZK21-29).

**Institutional Review Board Statement:** Not applicable.

**Informed Consent Statement:** Not applicable.

**Data Availability Statement:** The data presented in this study are available on request from the corresponding author.

**Conflicts of Interest:** The authors declare no conflict of interest. The funders had no role in the design of the study; in the collection, analyses, or interpretation of data; in the writing of the manuscript, or in the decision to publish the results.

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
