# Peer review of "Drone Based RGBT Tracking with Dual-Feature Aggregation Network"

_drones, doi:10.3390/drones7090585_

Round 1

Reviewer 1 Report

Overall Evaluation:

The paper introduces a dual-feature aggregation network for fusing visible and infrared information for object tracking. However, there are areas for improvement in certain aspects. Here are three review points for this manuscript:

Innovation Description:

1.     Although the paper mentions the dual-feature aggregation network, it does not provide sufficient details about the attention mechanism.It also lacks an explanation of how the attention mechanism establishes connections between different modalities.

Comparison Analysis:

2. When introducing the dual-feature aggregation network, a thorough comparison with traditional visible-infrared fusion methods is advisable. This would demonstrate the new approach's performance, efficiency, and robustness advantages.

Experimental Validation:

3. The algorithm comparison in the paper seems somewhat limited. Providing a comprehensive comparison on multiple benchmark datasets would better.

In conclusion, the innovation presented in the paper is commendable, yet there is room for further enhancement in the description of the innovative mechanism, comparison analysis, and experimental validation to elevate the quality and comprehensibility of the manuscript.

Reviewer 2 Report

This paper presented a novel method for RGBT object tracking based on UAV image. Overall, the structure of this paper is well organized, and the presentation is relatively clear. The idea is interesting and has potential. However, there are still a few problems that need to be carefully addressed. More specifically,

1. Visual-based tracking is widespread in robotics, autonomous driving, and intelligent transportation: an automated driving systems data acquisition and analytics platform, integrated inertial-lidar-based map matching localization for varying environments, hydro-3d: hybrid object detection and tracking for cooperative perception using 3d lidar. Thus, it is essential to include the above work and expand the applications. In addition, please explain the challenge of visual-based tracking compared to RGBT-based methods in detail.

2. As you mentioned, Transformer-based methods have achieved great success in computer vision. However, it may depend on the scene. For example, in yolov5-tassel: detecting tassels in rgb uav imagery with improved yolov5 based on transfer learning, the CNN-based method could achieve higher accuracy compared to the Transformer-based methods for remote sensing image detection. It would be better to include the above case and explain why.

3. Please make the font of the horizontal and vertical coordinates in Figure 7 smaller.

4. For section 4.3, it would be better to introduce different comparative methods briefly.

5. In Table 2, besides FPS, it would be better to include the parameter size of each model as well.

Reviewer 3 Report

Comments attached

Round 2

Reviewer 3 Report

The Authors made minimal but sufficient adjustments.

Line 40 - Delete "see"